# Diversity spurs diversification in ecological communities

Vincent Calcagno[1], Philippe Jarne[2], Michel Loreau[3], Nicolas Mouquet[4] & Patrice David[2]

Diversity is a fundamental, yet threatened, property of ecological systems. The idea that diversity can itself favour diversification, in an autocatalytic process, is very appealing but remains controversial. Here, we study a generalized model of ecological communities and investigate how the level of initial diversity influences the possibility of evolutionary diversification. We show that even simple models of intra- and inter-specific ecological interactions can predict a positive effect of diversity on diversification: adaptive radiations may require a threshold number of species before kicking-off. We call this phenomenon DDAR (diversity-dependent adaptive radiations) and identify mathematically two distinct pathways connecting diversity to diversification, involving character displacement and the positive diversity-productivity relationship. Our results may explain observed delays in adaptive radiations at the macroscale and diversification patterns reported in experimental microbial communities, and shed new light on the dynamics of ecological diversity, the diversity-dependence of diversification rates, and the consequences of biodiversity loss.

[1] Université Côte d'Azur, CNRS, INRA, ISA, Sophia Antipolis 06900, France. [2] CEFE UMR 5175, CNRS-Univ. of Montpellier-Univ. P. Valery Montp.-EPHE, Montpellier 34090, France. [3] Theoretical and Experimental Ecology Station, CNRS-Univ. Paul Sabatier, Moulis 09200, France. [4] MARBEC, CNRS-IFREMER-IRD-Univ. of Montpellier, Montpellier 34095, France. Correspondence and requests for materials should be addressed to V.C. (email: vincent.calcagno@inra.fr).

Diversity is a fundamental characteristic of ecological communities, that shows important variation around the globe, impacts many aspects of community and ecosystem functioning, and is globally threatened by human activities[1]. Deciphering the multifarious relationship between diversity, in particular species number, and ecosystem functioning and stability has constituted one of the most long-lasting research programs in ecology[2–4].

Given the importance and pervasiveness of diversity, evolutionary ecologists have put a lot of effort into understanding how diversity is generated in communities. For example, it has long been recognized that, when colonizing a habitat with vacant niches, a species may respond to this 'ecological opportunity'[5] by diversifying into several daughter species, each occupying different parts of ecological space. This phenomenon, called an adaptive radiation, is most spectacular in isolated ecosystems such as remote archipelagos, where species colonization is so rare that most, if not all, of species diversity can result from the diversification of an ancestral species[5,6]. Theoretical models of frequency-dependent ecological interactions have successfully been used to investigate how adaptive radiations could be an outcome of the mutation-selection process[7–10]. We now have a good understanding of how phenotypic evolution, by continually reshaping the fitness landscape, can under certain conditions cause species to split into two diverging lineages. This process can repeat itself several times, effectively leading to an adaptive radiation, as it was shown to occur in several models of ecological interactions[10].

Among the various consequences of diversity, one particularly popular and controversial hypothesis is that diversity itself promotes diversification, diversity thus driving its own genesis[11–13]. However, whilst the ecological and functional implications of diversity have received considerable attention, its impact on the possibility of evolutionary diversification is still, surprisingly, poorly understood. Evolutionary theory classically investigates whether one species, in an isolated system, evolves to a fitness maximum, at which selection is stabilizing, or to a fitness minimum, at which selection is diversifying. In the former case, the species is said to have attained an evolutionary stable strategy (ESS) and no further change occurs. In the latter case, the species attains an evolutionary unstable strategy and may start to diversify[7,14–16]. However, the evolutionary stability of diverse assemblages is rarely studied in detail[16,17]. This is unfortunate, considering that most adaptive radiations occur in partly connected systems, combining potentially multiple species colonization events and evolutionary diversification[18]. It is therefore important to understand how the immigration of one or several species could modify the evolutionary stability of ecological assemblages, and therefore control the occurrence of adaptive radiations[19,20].

In this study, we investigate the connection between diversity (initial number of species) and evolutionary stability (the possibility of adaptive diversification) in theoretical ecological communities. We consider a generalized model describing the temporal dynamics of species densities as governed by intra- and inter-specific ecological interactions (Methods section). The model can encompass several classical community ecology models. Species are characterized by their trait value $x$, and trait values control the intrinsic growth rates ($r$) and carrying capacities ($k$) of species, as well as the type and strength of pairwise species interactions ($a$). Different ecological scenarios correspond to different shapes of functions $r$, $k$ and $a$. For numerical applications, we consider three contrasted scenarios representative of standard models of species coexistence, specifically the niche scenario (symmetric competition[7,21–23]); the body-size scenario (asymmetric competition[10,24]) and the life-history (LH) trade-off scenario (competition-colonization dynamics[8,25,26]). An overview of the functions corresponding to the three scenarios is provided in Fig. 1 (see Methods section for further details).

For each scenario, we determine the possibility of adaptive radiation by studying the adaptive dynamics of species trait values, as a function of the initial number of species (Methods section). In all three scenarios, we find that the likelihood of adaptive radiation increases with the initial number of species. We show that, over large parts of parameter space, adaptive radiation can occur only if a sufficient initial number of species are present, so that adaptive radiations can be triggered by the arrival of one or more species, a phenomenon we call diversity-dependent adaptive radiation (DDAR). We analyse mathematically the selective causes of this phenomenon and argue that it can explain diversification patterns observed at different spatial and temporal scales. We also discuss the implications of DDAR for the dynamics of diversity, especially in a context of habitat loss.

## Results

**Diversity-dependent adaptive radiations (DDAR).** We ran simulations of the stochastic mutation-selection process under the three ecological scenarios. We systematically varied the initial

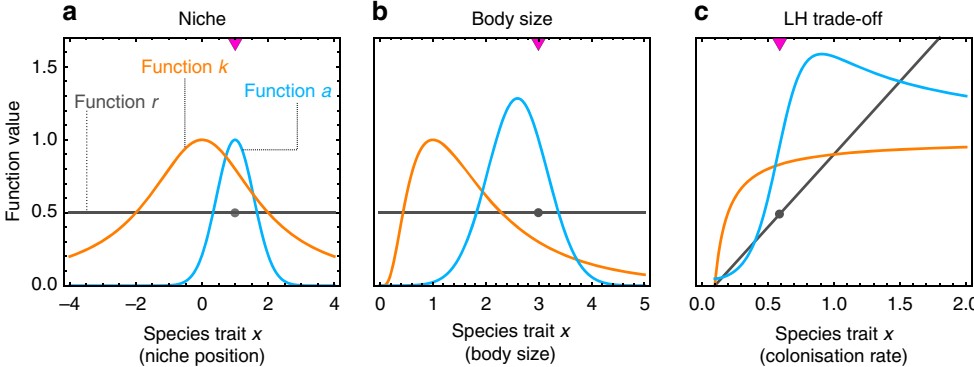

**Figure 1 | Specific functions corresponding to the three ecological scenarios.** Functions $r$, $k$ and $a$ are shown for the niche (**a**), body-size (**b**), and life-history (LH) trade-off (**c**) scenarios. For simplicity we depicted a situation with one resident species (with trait value $x_i$; purple triangles and dots). Functions $r$ and $k$ are the intrinsic growth-rate and carrying capacity of a species, respectively, as a function of its trait value. Note that there is an intermediate $k$ optimum in the niche and body-size scenarios, but not in the LH-trade-off scenario. For function $a$ we represented $a(x_i, x_j)$, that is, the impact that the resident species has on other species depending on their trait values. Note that $a$ is symmetric and less than one in the niche scenario, but asymmetric and possibly greater than one in the body-size and LH-trade-off scenarios.

number of species between one and five. In simulations started with one species, the species evolved a trait value that, depending on ecological scenario, either maximized carrying capacity (niche), was slightly off maximum carrying capacity (body-size) or conferred very low carrying capacity (LH trade-off; Fig. 2). Following this episode of directional evolution, the species could diversify several times as in a classical adaptive radiation[7,9,10]. However this did not occur for all ecological scenarios and parameter values. In the niche and body-size scenarios, it occurred only for some parameter combinations and in the LH trade-off scenario it never occurred, regardless of parameters. Instead, the species attained a fitness maximum or evolutionary stable strategy (ESS), precluding diversification (Fig. 2).

In those situations not conducive to adaptive radiation, we observed, in all three scenarios, that evolutionary diversification readily occurred provided sufficient initial diversity (number of species) was present. In other words, adaptive radiation, when impossible starting with one species, often became possible starting from two, three or more species (Fig. 2). Adaptive radiation can thus require some initial diversity, a phenomenon we call (diversity-dependent adaptive radiation) DDAR.

To evaluate the prevalence of DDAR, we used adaptive dynamics methods and numerical continuation, continuously varying two key model parameters in each ecological scenario (Methods section). For diversity levels between one and five, we tracked the location of evolutionary attractors and the changes in their evolutionary stability. Loss of evolutionary stability (bifurcation from ESS to branching point) occurs when one species in the community evolves to a fitness minimum, and thus no longer prevents invasion by nearby trait values (Fig. 2). This represents the disappearance of what is called limiting similarity in the context of the niche scenario, or the niche shadow in the context of LH trade-offs[27].

We were thus able to compute the likelihood of diversification (defined as the fraction of parameter space in which diversification can occur), as a function of initial diversity. We found that diversification likelihood increased steadily with diversity, and approached one, in all ecological scenarios: more diverse communities were less likely to be evolutionary stable (Fig. 3a–c). As adaptive radiation requires at least one species in the community to lose evolutionary stability, this pattern might simply reflect the fact that the more species, the greater the chance that one loses evolutionary stability. To account for this effect, we compared our results to the null hypothesis in which the per-species diversification probability stays constant, so that the likelihood of diversification increases geometrically with the number of species: more precisely, if diversification probability is $\gamma$ for one species, it should be $1 - (1 - \gamma)^s$ for $s$ species. As is visible in Fig. 3, the observed increase in diversification likelihood always exceeded this null hypothesis. An extreme example is the LH trade-off scenario, for which diversification likelihood jumped from zero to one between one and three species (Fig. 3). DDAR were thus the only form of radiation in this scenario.

For each parameter combination, we further computed the minimal initial diversity that was necessary to observe adaptive radiations, and identified boundaries in parameter space at which this required diversity level changed. This revealed that parameter space regions conducive to adaptive radiation gradually expand with initial diversity (Fig. 3). The conditions favouring diversification were qualitatively similar to those in the one-species case, but they were less stringent in multi-species communities. For instance, in the niche scenario, diversification is favoured by a smaller width of the competition function, at all diversity levels. But whereas with one species adaptive radiation is possible only if the competition kernel is narrower than the resource function, that is, $s_a < s_k$ (ref. 7), with two species adaptive radiation is possible even if $s_a$ is moderately larger than $s_k$, and with three the constraint on $s_a$ is essentially suppressed (Fig. 3). Similarly, in the body-size scenario, diversification is favoured by a narrower

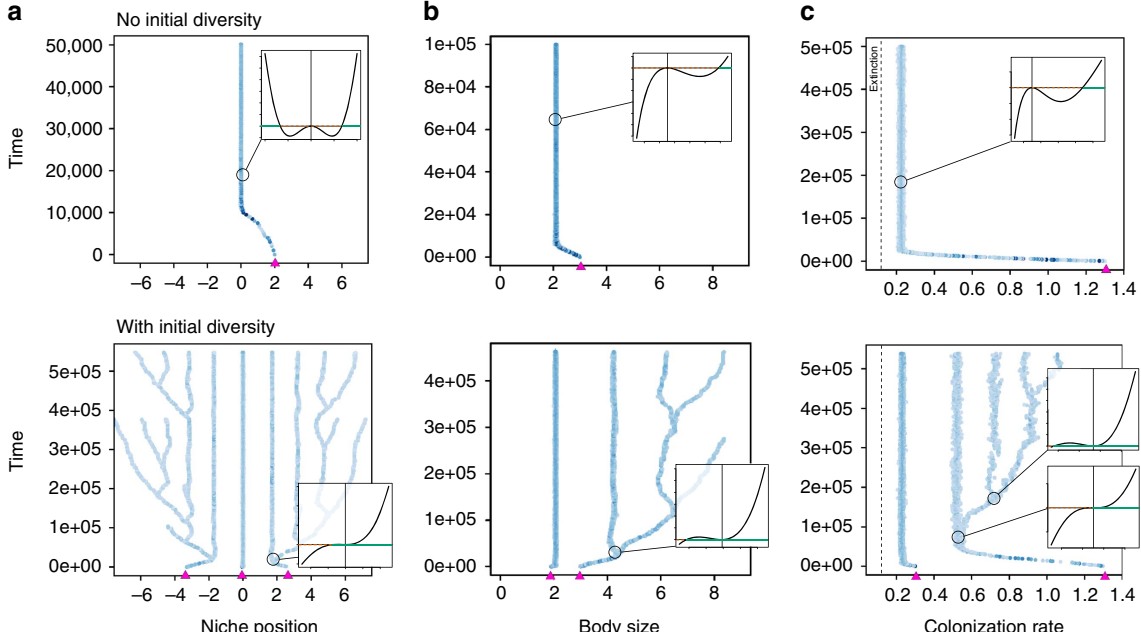

**Figure 2 | Instances of diversity-dependent adaptive radiation (DDAR) in evolutionary simulations.** Stochastic simulations are shown for the three ecological scenarios (**a–c**; Fig. 1). Trait values are plotted through evolutionary time, darker shades of blue indicate greater relative abundance. In the three panels, adaptive radiation did not occur if starting with only one species (top), but did occur, for the same parameters, if starting with more than one species (bottom). The initial species and their trait values are shown as purple triangles on the $x$-axes. Fitness landscapes are shown as inserts. The trait value of the focal species (circle) is located by the vertical line, and the invasion fitness of mutants around this value is plotted (Methods section). The horizontal line corresponds to zero fitness; trait values with positive fitness (solid line) can invade, others cannot (dashed line).

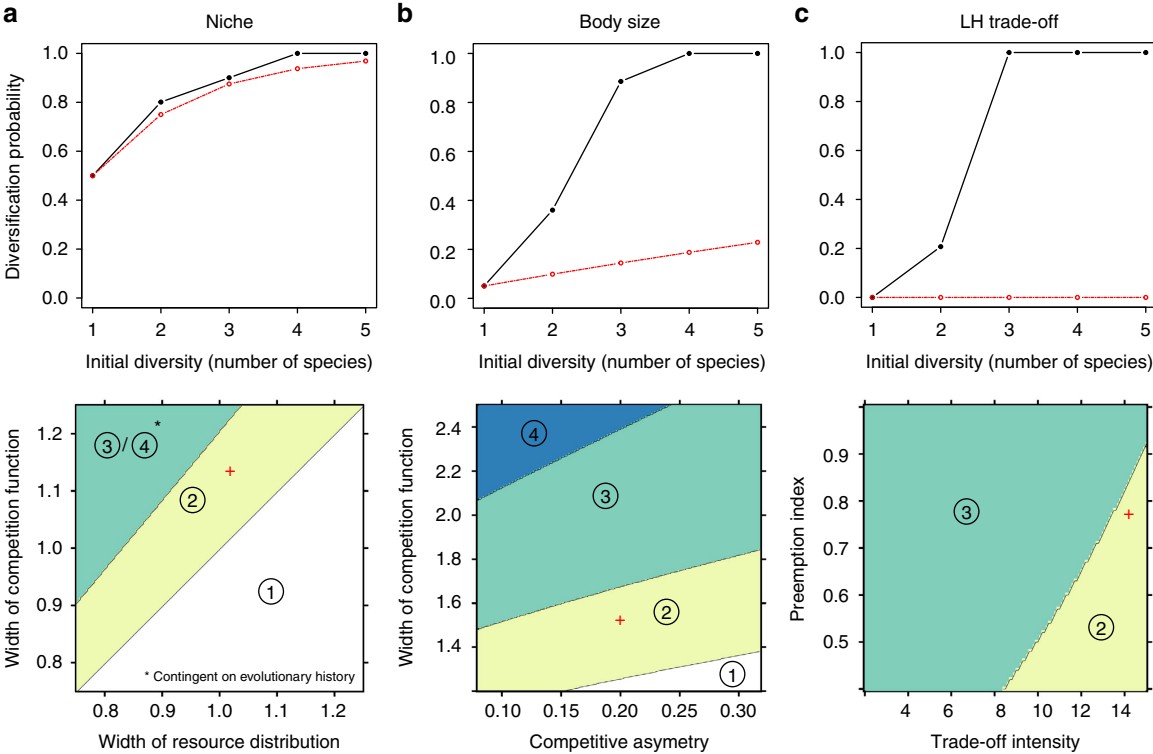

**Figure 3 | Diversification likelihood increases with diversity.** The likelihood of adaptive radiation (fraction of parameter space conducive to adaptive radiation) is shown as a function of initial diversity under the three ecological scenarios (**a–c**; top). Red dotted lines represent the null expectation if the per-species probability of diversification remained constant regardless of initial diversity. Also shown is the minimum initial diversity needed to observe an adaptive radiation, as a function of parameter values, under the three ecological scenarios (**a–c**; bottom). '1' indicates parameter combinations such that one-species evolution results in a classical radiation, '2' those such that adaptive radiation is impossible starting from one species, but is possible starting with two species or more (DDAR), and so on. In the niche scenario, there were alternative evolutionary attractors for three and four species, that differed in evolutionary stability. Hence for a given parameter combination, the minimum initial diversity further depends on historical contingencies, such as the initial trait values of species and the sequence of mutations that occur. Red crosses indicate the parameter combinations used in Fig. 2.

competition function and a stronger competitive asymmetry, but conditions become less stringent as diversity increases (Fig. 3). In the LH trade-off scenario, diversification is favoured by a greater trade-off intensity and a lower competitive preemption. Treating initial diversity as a parameter, the prediction common to all three scenarios is that diversification is favoured by greater diversity.

Apart from the minimum level of diversity needed to trigger the initial burst of diversification, DDAR are similar to classical adaptive radiations: diversity builds up in the community, at a pace that gradually slows down until the number of species reaches some plateau (Fig. 4a). The eventual stop of diversification can be explained by classical adaptive radiation theory: as the number of species increases, there is a decline in the diversification rate and an concomitant increase in the rate of extinction (Fig. 4b). Both processes can be attributed to the decrease in the average population density of species, caused by the progressive filling of niche space (Fig. 4c). The main difference between DDAR and classical adaptive radiation is thus the existence of a (potentially long) lagtime before the onset of diversification, corresponding to the time required for immigration processes to bring in sufficient initial diversity (Fig. 4a).

**Selective causes of DDAR.** Although the numerical computation of fitness landscape curvature suffices to determine changes in evolutionary stability, it does not inform on the selective factors underlying the observed changes. To get general insights into the causal link between diversity and diversification in the initial stages of adaptive radiations, we analysed the curvature of the fitness landscape mathematically, considering a generalized evolutionary attractor in the context of equation (2) (Methods section). We derived the following condition for adaptive radiation to be possible in a community of $s$ species:

$$\max_{i \in (1,s)} (H_i - I_i B) > 0 \qquad (1)$$

where

$$H_i = \frac{\mathrm{d}^2 k(x_i^*)}{\mathrm{d}x_i^2} \quad (1.1) \quad I_i = \frac{\partial^2 \sum_j p_j^* a(x_i^*, x_j^*)}{\partial x_i^2} \quad (1.2)$$

$$B = \sum_j n_j^* \quad (1.3)$$

Asterisks indicate that quantities are evaluated after coevolution, at an evolutionary singular point. If the condition is verified, at least one species is at a fitness minimum and the species assemblage is not evolutionary stable. Otherwise, the assemblage represents a (local) fitness maximum and diversification is impossible given the current number of species.

Equation (1) generalizes several earlier diversification conditions[10]. For instance, under the niche scenario with only one species, it reduces to the well-known criterion that the carrying capacity function should be broader than the competition function, that is, $s_k > s_a$[7]. Equation (1) shows that the evolutionary stability of ecological communities can be decomposed into three components:

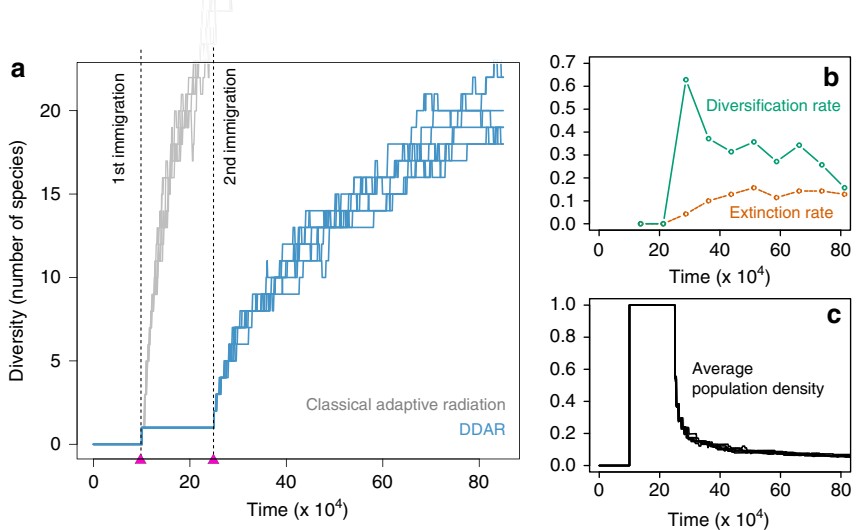

**Figure 4 | Impact of DDAR on the dynamics of diversity.** The number of species is shown as a function of time (**a**), in simulations starting from an empty community, under the niche scenario. Two sequential species colonizations were assumed (arrows). In a classical radiation (niche width $s_a = 0.9$), diversification starts after the first colonization (grey), whereas in the case of DDAR ($s_a = 1.1$) it cannot start until the second colonization (blue). Note that in this case the DDAR curve saturates faster as the competition function is broader, and fewer species can be packed. Seven replicate simulations are shown in both cases. (**b**) shows the average diversification (solid line) and extinction (dashed line) rates as a function of time in the case of DDAR. (**c**) shows the average population density as a function of time in the case of DDAR.

$H_i$ represents selection to match the habitat. An individual is selected to perform well in its habitat, which in this class of models amounts to maximizing $k$. This optimization force is frequency- and density-independent.

$I_i$ represents frequency-dependent selection on inter-individual interactions. These include both intra- and inter- specific interactions, weighted by their relative frequencies ($p_j^*$ and $1 - p_j^*$, respectively). An individual is selected to minimize the average impact other individuals exert on it (hence the minus sign in equation (1)).

$B$ is total community biomass, that is, the summed abundance of all species. It affects all species similarly, by governing the relative importance of $I_i$. As total biomass increases, $I_i$ contributes proportionally more to selection. This is because the greater total density, the more frequent inter-individual interactions. $B$ expresses the density-dependent nature of selection.

All three components varied with the number of species in our numerical experiments (Supplementary Fig. 1). DDAR indicate that equation (1) was easier to verify as $s$ increased. To dissect the underlying causes, we computed, for each ecological scenario, parameter combination, and diversity level, the value of the three components ($H_i$, $I_i$ and $B$) individually. We were thus able to determine the role played by each selection component in the loss of evolutionary stability, for each bifurcation from ESS to diversification (Methods section). We did this separately for each type of evolutionary bifurcation, that is, those occuring between $s = 1$ and $s = 2$ ('2' regions in Fig. 3), between $s = 2$ and $s = 3$ ('3' regions in Fig. 3), and so on. Results are summarized in Fig. 5a–c.

At low diversity levels the picture was similar in all ecological scenarios: evolutionary bifurcations were caused by variations in $H_i$ and, to a lesser extent, $B$, while variations in $I_i$ counteracted the onset of diversification (Fig. 5a–c). This can be understood in terms of two general ecological principles: character displacement and diversity-productivity relationships. Considering first the niche and body-size scenarios, as species diversity increases some species are pushed away from the monospecific attractor, which

is close to the resource optimum. As a result of this process, called character displacement[5,21], species tend to exploit more peripheral resources (Fig. 2a,b). As such species approach the inflexion point of the resource function, $H_i$ is much less negative, and possibly positive, so that its stabilizing effect weakens or vanishes. Second, a generic property of resource competition models is that total community biomass increases with the number of species, generating a positive diversity-productivity (or diversity-biomass) relationship, a pattern often observed in natural ecosystems[1]. The reason is that more species collectively exploit a greater range of available resources (niche complementarity). Hence $B$ steadily increases with $s$ (Supplementary Fig. 1). This strengthens the selective impact of $I_i$ (equation (1)), which is disruptive in monospecific communities. Hence, two processes can individually trigger DDAR: (i) character displacement away from the resource optimum, and (ii) the positive diversity-productivity relationship.

The same conclusions hold for the LH trade-off scenario (Fig. 5a–c), though for quite different biological reasons. First, it is not adoption of peripheral resources that causes relaxed stabilizing selection from $H_i$. Indeed, single species do not evolve to maximize $k$, but on the contrary adopt low $k$ values (low colonization rates; Fig. 2c). Function $k$, as it arises from patch dynamics[25], has no intermediate maximum or inflexion point, and $H_i$ is always negative (Fig. 1c). In multispecies assemblages, some species adopt higher colonization rates, and thus higher carrying capacity, to compensate for their competitive disadvantage (Fig. 2c). As function $k$ gets flatter as trait value increases (Fig. 1c), the end result is the same as in previous scenarios: $H_i$ becomes less stabilizing for those species pushed away from the monospecific attractor. Second, a positive diversity-productivity relationship is not expected for arbitrary species assemblages in the LH trade-off scenario, owing to the strongly interferential nature of competition and lack of niche complementarity. However, we found such a positive diversity-productivity relationship for coevolved species assemblages (that is, in terms of $n^*$; Supplementary Fig. 1), which are the ones that matter to control diversification (equation (1)).

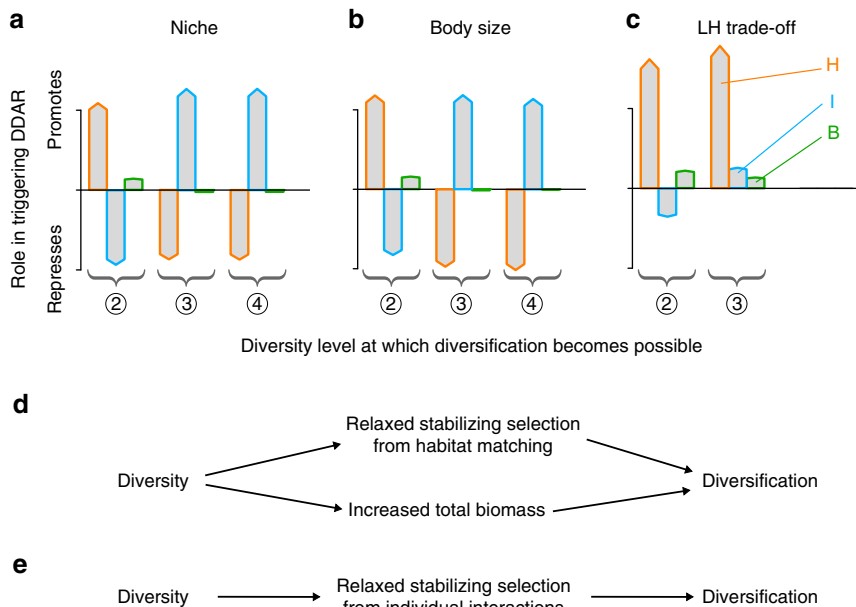

**Figure 5 | Selective causes of DDAR.** Vertical bars represent, under the three ecological scenarios (**a–c**), the relative contribution (positive or negative) of each component ($H_i$, $I_i$ and $B$; equation (1)) in causing bifurcations from ESS to evolutionary diversification as initial diversity increases. We showed separately bifurcations occurring at each diversity level, that is, those that occurred between one and two species, between two and three species, and between three and four species (corresponding to different regions in parameter space; Fig. 3). Synthetic diagrams summarize how diversity positively impacts diversification, at low diversity levels (**d**) and at high-diversity levels (**e**).

At higher diversity levels this picture could be significantly altered. In all three scenarios, the variation in $I_i$ reversed, and became favourable to diversification (Fig. 5a–c). In the niche and body-size scenarios, intraspecific interactions are disruptive, as similar individuals exploit similar resources and can escape competition by diverging in trait space. This is the main driving force of classical adaptive radiations[7,10]. However, with two or more species, this disruptive effect of intraspecific competition is greatly diluted, as a large fraction of interactions (or even most of them, for relatively rare, peripheral species) are heterospecific. The latter are stabilizing, as individuals are selected to avoid engaging in competition with other species. Because of this dilution effect, the value of $I_i$ in equation (1) switched from negative (disruptive) in monospecific communities to positive (stabilizing) in multi-species communities (Supplementary Fig. 1). As diversity increases further, however, species can be so different in trait space that selection on interactions with them, albeit positive, weakens considerably (as we reach the tails of function $a$; Fig. 1a,b). As a result $I_i$ decreases again, favouring diversification (Fig. 5a–c).

A similar reversal of the effect of $I_i$ occurred in the LH trade-off scenario, while $H_i$ and $B$ otherwise retained their effects. But in the niche and body-size scenarios, further changes occurred. First, the variation in $H_i$ also switched sign (Fig. 5a–b). The interpretation is similar as for $I_i$: peripheral species are displaced to resource portions, where $H_i$ becomes positive and thus is stabilizing. As diversity increases further, they are pushed far into the tails of function $k$, and $H_i$, though still positive, weakens (Fig. 1). The roles of $H_i$ and $I_i$ are thus effectively reversed. Second, component $B$ loses its relative importance (Fig. 5a–b). This echoes the saturating shape of the diversity-productivity relationship: increases in total biomass were most pronounced at low species numbers, and then slowed down (Supplementary Fig. 1).

From these analyses of the selection components, we can thus distinguish two causal pathways connecting diversity to diversification, and explaining the DDAR phenomenon. First, at low diversity levels, the positive effect of diversity on diversification was primarily caused by density- and frequency-independent selection on habitat matching and by the positive diversity-productivity relationship (Fig. 5d). Second, at high-diversity levels, the diversification-promoting effect of diversity was primarily caused by frequency-dependent selection on individual interactions (Fig. 5e). The two pathways operated simultaneously in the LH trade-off scenario at medium diversity levels (Fig. 5c). All three components collaborated to suppress evolutionary stability, making DDAR inevitable (Fig. 3c).

## Discussion

Diversity has long been recognized as a key determinant of the ecological stability of ecosystems[1,4], but its implications for evolutionary stability (and evolutionary diversification) are much less understood. The idea that ecological diversity could in itself favour diversification is an intriguing, but difficult to prove, hypothesis. It is usually regarded as requiring complex conditions, involving processes such as niche construction, ecosystem engineering, cross-feeding[28] or propagating diversification across trophic levels[29]. Here, we established a theoretical connection between diversity and evolutionary stability and found that more diverse communities are, all else equal, less likely to be evolutionary stable and more likely to undergo evolutionary diversification, in basic models of ecological interactions. This generates a phenomenon we called diversity-dependent adaptive radiations (DDAR).

Classical adaptive radiation theory predicts that after the initial burst of diversity, diversification should gradually slow down, on the basis that ecological niches get filled and ecological opportunity declines[5]. This results in a negative relationship between diversity and diversification rate[30,31]. Such a property is observed in our model, as in many ecological models[15,21], where the number of species tends to saturate following an adaptive radiation (Fig. 2). However, in the case of DDAR, the rate of diversification undergoes an initial increase, and effectively follows a hump-shaped curve (Fig. 4b). Our results therefore

indicate that the same ecological theory that predicts a negative diversity-diversification relationship can also predict, in the very first stages of adaptive radiations, a positive diversity-diversification relationship. DDAR provides an alternative theoretical explanation to positive diversity-diversification relationships, involving only the shape of natural selection, without resorting to complex ecological scenarios or population genetics arguments[11,32].

The mechanisms causing DDAR were largely consistent across ecological scenarios, but depended on the diversity level. We identified two general causal pathways linking diversity to diversification, whose relative importance varied with diversity. Our condition for diversification (equation (1)) generalizes earlier results with important differences: selection on habitat matching is not necessarily stabilizing, selection on inter-individual interactions is not necessarily disruptive, and finally, total biomass plays a key role in determining the relative importance of the two. The latter effect has usually been overlooked in one-species studies, as total biomass is just that of the focal species and is often treated as a constant[7,23]. In multispecies systems however, total biomass commonly increases with diversity, a pattern known as the positive diversity-productivity relationship[1,33,34]. We have shown that this relationship between diversity and total biomass contributes to generating a positive impact of diversity on diversification, establishing an interesting connection between the evolutionary stability of communities and a fundamental functional property of ecological systems. Adaptive radiations are classically predicted to occur when there is an ecological opportunity (vacant niches) and strong competition for resources. In our modelling results, ecological opportunity did not increase with diversity, but DDAR can be partly explained by an argument based on competition: greater total abundance (component $B$) increased the intensity of competition, all else equal, which played some role in triggering diversification, especially at low diversity levels (Fig. 5). However this is only part of the picture, as changes in the distribution of trait values (character displacement) typically had greater importance, and in multi-species communities competition could sometimes hinder, rather than promote, diversification (Fig. 5 and Supplementary Fig. 1).

DDAR can provide a parsimonious explanation to empirical patterns at macro- and micro-scales. At the macroscale, clades sometimes fail to diversify despite apparent ecological opportunity[32,35–37]. Some adaptive radiations present an atypical lagtime before diversity starts to accumulate[38,39]. For example, Hautmann et al.[38] report that rediversification of benthic communities in the late Permian was unusually slow and argue that diversification was delayed for lack of competition and immigration in this system. One of the best documented adaptive radiations, Anolis lizards in the Antilles archipelago, also presents such a puzzling pattern: the first lineages colonizing Puerto Rico did not cause an adaptive radiation, in contrast to other islands[36,37]. It is only the third[37], or perhaps the second[36] colonizing lineage that diversified into several species (Fig. 6a). We suggest that could simply be an instance of DDAR, even though other explanations involving environmental or geographical contingencies might be conceived. At the micro-scale, experimental studies of microbial adaptive radiations reported that the addition of a competitor could trigger and hasten evolutionary diversification[13,20]. This was attributed to an increase in the effective amount of resource competition, which is commonly thought to promote diversification. This is consistent with our finding that an increase in total biomass (component $B$) can in itself favour diversification, even though this is only one of the different ways through which competitive interactions impact diversification (Fig. 5). A recent study explicitly manipulated

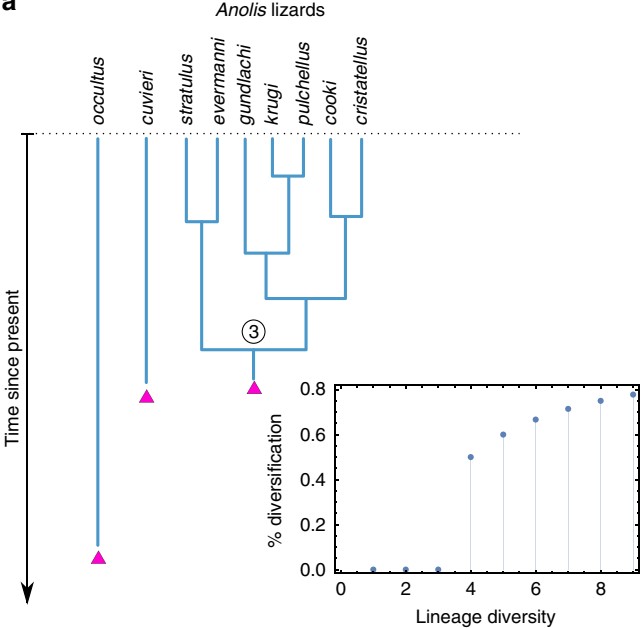

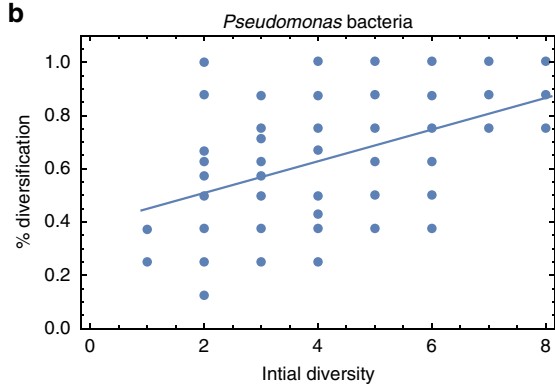

**Figure 6 | DDAR can explain patterns at the macro and micro scales.** The adaptive radiation of Anolis lizards in Puerto Rico (**a**) presents an initial lag. Under the maximum clade credibility phylogenetic tree[37], diversification did not start until the third colonization event or possibly, depending on species set and molecular markers, the second[36] (Supplementary Fig. 2). Data from ref. 37. Experimental evolution of the bacteria Pseudomonas fluorescens F113 (**b**) indicates that greater initial diversity (number of strains) promotes evolutionary diversification (fraction of evolved genotypes). The line represents a significant linear regression on diversity ($P = 0.0013$). Data from[40].

initial functional diversity and investigated its consequences on diversification in a microbial system[40]. As a reanalysis of their data (in terms of species number rather than functional diversity) indicates, a greater initial species richness promoted diversification, at diversity levels similar to those in our models (Fig. 6b). The authors further analysed the resource use profile of new genotypes and found that diversification was associated with a shift to scarcer, peripheral, resources, which is consistent with the first causal pathway we identified (Fig. 5d).

Our findings echo recent developments in evolutionary theory showing that complexity, defined as phenotype dimensionality[23,41,42] or overlapping ecological and evolutionary timescales[43], can affect the evolutionary stability of a species and promote evolutionary diversification. A general picture is thus emerging: evolutionary diversification is more likely in complex ecological ecosystems, and the conditions permitting

adaptive radiation could be much broader than previously thought. More importantly, a positive impact of diversity on diversification would have uniquely important consequences for our understanding of the dynamics of diversity and ecosystem development. It implies the possibility of hysteresis effects in the dynamics of diversity, reminiscent of Allee-effects in population biology[44]: species loss might not only compromise ecosystem functioning in the short term, but also impair the capacity of ecosystems to restore functions in the long term, once diversity has fallen below some threshold level. Such evolutionary debts[40] could prove especially problematic in naturally or anthropogenically isolated ecosystems.

## Methods

**Model.** We consider a generalized model describing the ecological dynamics of $s$ species interacting in a community. The model encompasses most particular models of adaptive radiation that have been studied so far, and does not assume particular functional forms, allowing us to derive general insights[43,45]. The abundance $n_i$ of a particular species $i$ changes through time according to

$$\frac{1}{n_i}\frac{dn_i}{dt} = r_i g\left(\frac{k_i}{\sum_{j=1}^{s} n_j a_{i,j}}\right) \text{ for } i \text{ in } 1 \ldots s \qquad (2)$$

The growth function $g(\cdot)$ should be increasing and satisfy $g(1) = 0$. By far the most common choice is $g(\cdot) = 1 - 1/\cdot$, corresponding to Lotka–Volterra equations. A generalization is $g(\cdot) = (1 - 1/\cdot^\beta)/\beta$, in which case $\beta = 1$ yields Lotka–Volterra equations and $\beta = 0$ Gompertz equations[46]. For plant communities, an alternative form is $g(\cdot) = \cdot - 1$, following from Wit's growth model and competitive lotteries[47,48]. In all cases, the argument of $g$ is the ratio between the intrinsic growth potential of a species (parameter $k_i$ at the numerator) over the population density effectively experienced by the species ('perceived crowding'; denominator). The latter is a weighted summation over all individuals in the community, in which individuals of species $j$ have weight $a_{i,j}$, capturing the net impact of species $j$ on species $i$. When numerator and denominator balance out, the species is at an equilibrium abundance. It is common practice to scale parameters so that $a_{i,i} = 1$, which makes $k_i$ the carrying capacity of species $i$, that is, its steady-state abundance when growing alone. Parameter $r_i$ governs the timescale of species dynamics, as notably influenced by generation time. Each species is characterized by some functional trait $x_i$ (for example, body size), which for simplicity is taken to be scalar. Species traits may affect all parameters of the ecological interactions, and so we let $r_i = r(x_i)$, $k_i = k(x_i)$ and $a_{i,j} = a(x_i, x_j)$, with $r$, $k$ and $a$ sufficiently smooth functions.

**Ecological scenarios.** The shapes of functions $r, k$ and $a$ determine the type of ecological interactions at play in the community. Although our generalized model does not put any restriction on the form of interactions, for simulations and numerical investigations we considered three contrasted special cases that have received most attention as adaptive radiation models: (a) the niche scenario, (b) the body-size scenario and (c) the life-history (LH) trade-off scenario (Fig. 1). A full comparison of the three scenarios is provided in Supplementary Table 1. The niche scenario is probably the most classical scenario for adaptive radiations. It is a simple description of how competition for a continuous resource spectrum can result in speciation[7,21–23]. Trait $x$ is a key functional trait describing niche position, such as beak size for granivorous birds feeding on a range of seed sizes. Resources are assumed to be unimodally distributed, and species compete less intensely when they overlap less in the range of resources they utilize, making $a$ an even function of trait difference $x_i - x_j$ (Fig. 1a). Under the body-size scenario, trait $x$ represents average body-size and larger species are assumed to have a competitive edge over smaller ones[10,24]. This breaks the symmetry of $a$ with respect to the trait difference and allows it to take values greater than one, which means that inter-specific impact can exceed intraspecific impact (Fig. 1b). Finally, the LH trade-off scenario considers interactions between species that are dominant over different parts of their life cycle[9]. As an example of this scenario, we consider the classical competition-colonization trade-off: species that are dominant at exploiting local resource patches are less efficient at dispersing and colonizing open portions of the habitat[8,25,26]. In this case, $n_i$ represents the fraction of available spatial localities (sites) occupied by species $i$ in the habitat, and $x_i$ is the rate at which it colonizes new patches. The LH trade-off scenario represents an asymmetric form of ecological interactions like the body-size senario, but differs considerably in that function $k$ does not have an intermediate maximum, and function $a$ never shrinks to zero on one side[49] (Fig. 1c).

For numerical investigations and simulations, we considered three special cases of the generalized model (2), with a Lotka–Volterra $g$ function and contrasted functional forms for $r$, $k$ and $a$, as illustrated in Fig. 1. For the niche and body-size scenarios, the original assumption that $k$ and $a$ are both gaussian can result in undesirable or non robust model behaviours and other functions wih the same qualitative properties are increasingly preferred[17,23,50]. We thus considered several choices for the functional forms: generalized normal functions (varying exponent

from gaussian to quartic), boosted gaussian functions (Gaussian plus a constant), or Lorentzian (Cauchy) functions. As we obtained similar patterns for these different choices, we report results for the functional forms that are simplest and closest to the original modelling choices: (i) in the niche scenario, we used a Lorentzian function for the carrying capacity $k$ and the usual gaussian function for the competition function $a$; (ii) the body-size scenario, we used a log-normal function for $k$ and the usual gaussian function for $a$; and (iii) in the LH trade-off scenario, we retained the same assumptions as in the original model[26]. Full details on the model equations, and the formulation of LH trade-off scenario as an instance of equation (2), are presented in Supplementary Note 1. We note that the methodology could be directly applied to an even broader set of ecological situations, as our model can also generalize other forms of ecological trade-offs (for example, ref. 48) or mutualistic and trophic interactions (for example, the foodweb model by[51] can be formulated as a special case of (2)).

**Stochastic simulations.** We simulated ecological dynamics under the three ecological scenarios, allowing the trait values $x$ of species to evolve through a stochastic clonal mutation-selection process. Equation (2) was integrated using an adaptive fourth-order Runge–Kutta algorithm, incorporating a mutation rate (in the range $10^{-3}$ to $10^{-2}$), so that at each integration step there was a small chance that a mutation occured. If so, we selected randomly, in proportion to abundance, which genotype was affected, and a small density ($10^{-3}$) of individuals were then attributed a new trait value, equal to the original trait plus a random deviate taken from a gaussian distribution with mean zero and variance $10^{-6}$. To avoid atto-fox effects genotypes which abundance fell below the initial abundance of novel mutants were considered extinct. The composition of the community (trait values, abundances) was recorded every 100 steps for the entire simulation time. The number of species at any time was computed as the number of distinct modes in the distribution of trait values. Extinction and diversification rates were then computed from changes in species number. All simulations were conducted in R[52].

**Numerical and mathematical analysis.** We used adaptive dynamics methods to study the selection gradient on traits $x_i$ generated from equation (2), under the assumption of ecological steady state. For each ecological scenario, we continously varied two key model parameters (Supplementary Table 2 for parameter ranges) and tracked the position of evolutionary equilibria (at which all selection gradients vanish) and corresponding species equilibrium abundances, using a numerical continuation approach, for diversity levels between one and five species. We determined the evolutionary stability of evolutionary attractors by numerically computing the curvature of the fitness landscape around every species. Finally, for all evolutionary bifurcations caused by a change in diversity, we computed the associated variation in the three selection components individually. We then calculated their relative importance in causing the observed evolutionary bifurcations, averaged them over all parameter combinations and scaled them so that absolute values add up to 100%. All mathematical calculations and the derivation of equation (1) from equation (2) are provided in Supplementary Note 2, and detailed analyses of evolutionary dynamics in the niche scenario are presented in Supplementary Note 3.

**Data availability.** The R code used for evolutionary simulations and analyses is available from the authors upon request. The data that underlie Fig. 6 are available from the authors of the corresponding publications upon request.

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

## Acknowledgements

We thank Luke Mahler and Alexandre Jousset for sharing data and Jonathan Losos for input on Anoles. V.C., P.J. and P.D. were funded by grant ANR-BIOADAPT-2012-AFFAIRS. M.L. was supported by the TULIP Laboratory of Excellence (ANR-10-LABX-41) and by the BIOSTASES Advanced Grant, funded by the European Research Council under the European Union's Horizon 2020 research and innovation programme (grant agreement No 666971).

## Author contributions

V.C. conceived the project in concertation with P.D., N.M., M.L. and P.J., V.C. performed the analyses and simulations, and wrote the manuscript with input from all authors.

## Additional information

**Competing interests:** The authors declare no competing financial interests.

