## [Peer Review File · Nature Communications]

Reviewers' comments:

Reviewer #1 (Remarks to the Author):

This report addresses broadly interesting questions about how and why biological diversification occurs. It focuses in particular on the phenomenon of adaptive radiation, which may account for most of biological diversity but is dramatically understudied from a theoretical perspective. Overall, I find the theoretical work presented here very interesting because it represents progress developing more sophisticated models of adaptive radiation and poses some interesting hypotheses that warrant further attention.

I am not a theoretician and I am not able to provide a detailed critique of the models developed in this report, particularly in the required review timeframe. My main criticism of this report is that the theoretical models it develops are largely detached from empirical evidence and at least one of the empirical examples that is used to support the model developed here is misinterpreted.

The authors interpret the Puerto Rican anole lizard adaptive radiation as supporting their model because it did not start to diversify impressively until two or more lineages were present on the island (lines 293-298, Fig 5). Unfortunately, this interpretation fails to recognize that Puerto Rico was connected to parts of Hispaniola until 15-20 mybp (and to parts of other island before that). Thus, the two older lineages on Puerto Rico are remnants of a lizard community that existed while Puerto Rico was part of a larger landmass and the anole radiation that Calcagno interpret as evidence for DDAR resulted simply from the fact that this endemic radiation could only evolve after the island of Puerto Rico attained geographic isolation from nearby landmasses. It appears as if this radiation occurred very shortly after Puerto Rico became isolated. I do not believe this example should be used to support the theoretical model of DDAR developed in this report because it does not suggest that radiations must attain some critical mass prior to the onset of diversification.

Reviewer #2 (Remarks to the Author):

This paper uses mathematical models to show that adaptive radiations are more likely to occur when they start not from a single ancestral species, but when there a more than one initial species in the ecosystem. These results are very interesting and biologically relevant, and they shed new light on an important macroevolutionary phenomenon. The study appears to be very well done, and I was impressed by the fact that the authors not only present phenomenological results (i.e., simulations), but also carefully derive a mechanistic explanation for the phenomena observed.

Nevertheless, I have a couple of concerns that the authors should address.

1. One big question that I think the authors need to at least discuss is that the phenomenon that they describe, DDAR, i.e., diversity-dependent adaptive radiations, must eventually become ineffective as the diversity increases in the evolving community. Or is diversity increasing without bounds in their communities? If not, diversity must saturate, which means that eventually, diversity cannot beget more diversity, so that DDAR is ineffective. If that is indeed the case, then the question becomes: what is the tipping point? When does the mechanistic explanation given in Figure 4 start to fail? And Why? I think these questions need to be answered at least to some extent if the DDAR phenomenon is to be well understood.

2. I don't really understand why in the niche model, diversification is possible in the presence of two initial species in cases where diversification is not possible if there is only one initial species (i.e., if

there is no branching in 1-species evolution). For example, in the Gaussian case (i.e., in the case where both the carrying capacity and the competition kernel are Gaussian functions), it is well known, and quite easy to see, that if there is no branching in 1-species evolution, it is indeed possible that two species can coexist. However, in that case there is no evolutionary equilibrium for evolution in the 2-species system. Instead, if the two coexisting species are allowed to evolve, they both simply evolve back to the singular point in the 1-species system, i.e., to the maximum of the carrying capacity. Based on this, I don't understand how the authors can observe diversification starting from two coexisting species. What am I missing? Is the above scenario of evolution of two coexisting species back to the ESS when the branching condition is not satisfied different in the quartic (non-Gaussian) case assumed by the authors? I think this needs some careful explanations.

Very minor points:

I. 67: why does (1) become a "Gompertz" equation for $\beta=0$?

I.128: "A bifurcation from ESS to diversification occurs when, after evolution, one of the species no longer prevents invasion by nearby trait values." This is a very vague statement. Are you referring to the bifurcation during which an ESS becomes an evolutionary branching point?

Reviewer #3 (Remarks to the Author):

This paper outlines a theoretical analysis suggesting that evolutionary diversification, like that associated with adaptive radiation, is more likely to occur when there is already some diversity present in a region. This hypothesis, which has sometimes been called the 'diversity begets diversity' hypothesis, is a controversial one, as it is typically assumed that diversification is most likely to occur when a single species gains access to a wide range of ecological opportunities.

The authors examine a simple, logistic-type model describing the population growth of a single species and how this can be impacted by the presence of other species. They show that, for at least three different ways of describing how competition occurs and impacts population density. The three models of competition are: (1) the 'niche model' describes competition for a range of resources; (2) the 'body size' model describes competition mediated through body size differences; and, (3) the 'life history' model describes a trade-off between competitive ability and colonization rate in a multi-patch environment). Their key result is that, in all three cases, evolutionary diversification in key trait values (resource use, body size, competition-colonization trade-off) is more likely to occur as the number of initial species present increases.

The causes of this relationship I find hard to parse, especially as I am not a theoretician. The authors attribute it to the relative importance of habitat matching, interspecific interactions, and biomass and how these impact so-called 'evolutionary stability' as the number of species increases. I'm confused on two issues. The first is 'evolutionary stability', which is not formally defined (or if it is, I missed it). I think it means, in the parlance of adaptive dynamics theory, the likelihood that a given population or collection of species can be invaded by a novel type. The second is the connection between the classic models of adaptive radiation – which sees diversification due to strong resource competition in the presence of abundant ecological opportunity (vacant niche space) is not made explicit. Is the effect of diversity on the propensity to diversify the result of changes to the strength of resource competition or the extent of ecological opportunity, or both? I think it is the latter, given the importance of habitat matching, total biomass, and species interactions, in their selection analysis. But I really am struggling to make sense of this section. To facilitate interpretation it might be useful to cast their models against the standard models of adaptive radiation, to highlight how they differ and the causal factors

driving the increased likelihood of diversification.

My last concern is that I am not entirely convinced that a diversity-begets-diversity scenario is that common in 'real' adaptive radiations. While the authors do make an effort to identify a couple of instances where this might occur (in Anoles and in laboratory experiments with *P. fluorescens*) the vast majority of examples in both the real world (see Schluter's seminal book, *The Ecology of Adaptive Radiation*) and in the lab (consult Kassen's book, *Experimental Evolution and the Nature of Biodiversity*) suggest that diversity-dependent adaptive radiation is not very common. The authors cite, for example, the paper by Bailey et al (2013) suggesting that competition can sometimes promote adaptive radiation as evidence to support their idea, but I think this is a mis-reading of their results. Competition did promote adaptive radiation in this system, but only when the competitor could not itself diversify. This effectively means that the strength of resource competition was stronger in the presence of a competitor, only because that competitor was already present (ie – one didn't have to wait for it to evolve *de novo*). Perhaps this is the point the authors are trying to make. If so, it would be useful to be clear about this from the start.

Finally, it might be useful to examine the dynamics of diversification in terms of the number of species through time more formally. Classic models of adaptive radiation predict this dynamic to be an S-shaped curve. How does this model change that prediction? Is the rate of diversification of a focal lineage faster, or is the total number of descendent species higher? Inspection of figure 2 suggests it is both but I would prefer to see a contrast between a classic model of adaptive radiation and these alternative formulations. Such an analysis would facilitate the interpretation of existing macroevolutionary data sets on diversification.

Despite these extensive comments, I think the model they have studied is intriguing and the results fairly compelling. My comments should be taken in the spirit of trying to improve clarity and to place their work into a context that can be more easily interpreted by those actively studying adaptive radiation in nature or in the lab.

Minor comments:

1. There are a number of odd uses of language throughout that I suspect are translation issues. Some I found: "vanish off" (line 27), "have spent continual efforts" (line 29), "seize this ecological opportunity" (is an anthropomorphism and attributes agency to a diversifying lineage, which is incorrect; line 33). Please revise for clarity.
2. The main point of the paper is only introduced in paragraph 3. Consider reorganizing and condensing paragraphs 1 and 2 to get to the main point faster. The link between diversity and community stability (paragraph 1) really isn't what this paper is about, for example, and is needlessly distracting.
3. n_i is introduced on line 101 but not referred to in equation 1, which caused me some confusion. Please clarify.

Response to referees

NOTE: in what follows all line, reference and figure numbers correspond to the revised manuscript.

Reviewer #1 (Remarks to the Author):

This report addresses broadly interesting questions about how and why biological diversification occurs. It focuses in particular on the phenomenon of adaptive radiation, which may account for most of biological diversity but is dramatically understudied from a theoretical perspective. Overall, I find the theoretical work presented here very interesting because it represents progress developing more sophisticated models of adaptive radiation and poses some interesting hypotheses that warrant further attention.

I am not a theoretician and I am not able to provide a detailed critique of the models developed in this report, particularly in the required review timeframe. My main criticism of this report is that the theoretical models it develops are largely detached from empirical evidence and at least one of the empirical examples that is used to support the model developed here is misinterpreted.

The authors interpret the Puerto Rican anole lizard adaptive radiation as supporting their model because it did not start to diversify impressively until two or more lineages were present on the island (lines 293-298, Fig 5). Unfortunately, this interpretation fails to recognize that Puerto Rico was connected to parts of Hispaniola until 15-20 mybp (and to parts of other island before that). Thus, the two older lineages on Puerto Rico are remnants of a lizard community that existed while Puerto Rico was part of a larger landmass and the anole radiation that Calcagno interpret as evidence for DDAR resulted simply from the fact that this endemic radiation could only evolve after the island of Puerto Rico attained geographic isolation from nearby landmasses. It appears as if this radiation occurred very shortly after Puerto Rico became isolated. I do not believe this example should be used to support the theoretical model of DDAR developed in this report because it does not suggest that radiations must attain some critical mass prior to the onset of diversification.

> We agree that this could provide an alternative explanation, one that invokes geological contingencies. However it appears that these geological considerations are not settled, as the geological history of the Antilles is quite complex and not fully resolved. For instance, this is not the interpretation retained by the authors of the specialized papers we cite on this matter (Prs Losos and Mahler). Moreover the reviewer's suggestion would raise additional questions, such as why most lineages found on Hispaniola are not on Puerto Rico, or why *A. occultus* is not a sister taxon to a Hispaniolan species, instead of being one of the deepest lineages in the tree. Nonetheless, we have added in the text that alternative explanations might also exist (lines 318-319). We chose to maintain Anoles as a potential empirical instance of DDAR though, as (i) we believe it helps readers relate our theoretical findings to actual ecological patterns, and (ii) alternative explanations to the phylogenetic pattern remain controversial. Concerning empirical examples in general, we have added one additional potential example from the fossil record (article by Hautmann et al. 2015; ref 44), in which the authors explicitly suggest there was a lagtime before the onset of diversification.

Reviewer #2 (Remarks to the Author):

This paper uses mathematical models to show that adaptive radiations are more likely to occur when they start not from a single ancestral species, but when there a more than one initial species in the ecosystem. These results are very interesting and biologically relevant, and they shed new light on an important macroevolutionary phenomenon. The study appears to be very well done, and I was

impressed by the fact that the authors not only present phenomenological results (i.e., simulations), but also carefully derive a mechanistic explanation for the phenomena observed.

Nevertheless, I have a couple of concerns that the authors should address.

1. One big question that I think the authors need to at least discuss is that the phenomenon that they describe, DDAR, i.e., diversity-dependent adaptive radiations, must eventually become ineffective as the diversity increases in the evolving community. Or is diversity increasing without bounds in their communities? If not, diversity must saturate, which means that eventually, diversity cannot beget more diversity, so that DDAR is ineffective. If that is indeed the case, then the question becomes: what is the tipping point? When does the mechanistic explanation given in Figure 4 start to fail? And Why? I think these questions need to be answered at least to some extent if the DDAR phenomenon is to be well understood.

> In this article we focus on the initial stage of adaptive radiations, when diversity is low (one or a few species), as we seek to understand the conditions favoring the onset of adaptive radiations. In the final stages of adaptive radiations, diversity eventually tends to saturate, as ecological niches are “filled”. This can be seen in the simulated radiations in Figure 2, and conforms to the standard adaptive radiation model; DDARs are no different from regular radiations in that respect. It is a point we did not address in the original manuscript, as early stages of adaptive radiations (diversification) and final stages (community filling) are usually modeled separately, using rather different approaches. Why the diversification-promoting effect we report at low diversity levels should eventually disappear or become dominated by other factors, so that diversity stops increasing and stabilizes on some equilibrium level is indeed an interesting issue.

We have addressed this point in the revised manuscript by explicitly considering the saturation of diversity following an adaptive radiation, and provide explanations of the pattern. This led us to add an entirely new Figure (Figure 4) and an additional paragraph in the Results section (lines 161-170). In the new Figure, the dynamics of species accumulation is plotted and one can visualize the slowing down of diversification mentioned by the Reviewer. To explain this pattern we also provide the diversification and extinction rates, and how they change in relation to the declining population of species (Fig. 4b-c). This indicates that the mechanisms we identified continue to operate in the late stages, but are superseded by other mechanisms related to population size. These mechanisms are not specific to DDAR and are the same that operate in classical adaptive radiations. We also discuss this aspect in greater detail in the Discussion (lines 276-285).

2. I don't really understand why in the niche model, diversification is possible in the presence of two initial species in cases where diversification is not possible if there is only one initial species (i.e., if there is no branching in 1-species evolution). For example, in the Gaussian case (i.e., in the case where both the carrying capacity and the competition kernel are Gaussian functions), it is well known, and quite easy to see, that if there is no branching in 1-species evolution, it is indeed possible that two species can coexist. However, in that case there is no evolutionary equilibrium for evolution in the 2-species system. Instead, if the two coexisting species are allowed to evolve, they both simply evolve back to the singular point in the 1-species system, i.e., to the maximum of the carrying capacity. Based on this, I don't understand how the authors can observe diversification starting from two coexisting species. What am I missing? Is the above scenario of evolution of two coexisting species back to the ESS when the branching condition is not satisfied different in the quartic (non-Gaussian) case assumed by the authors? I think this needs some careful explanations.

> The reviewer is right that in the niche model, for the particular Gaussian case (Gaussian form for both the competition and the carrying capacity functions), ESS for one species implies that two

species cannot coexist on an evolutionary timescale: the two species will collapse to the one-species singular point. In other words, evolution tends to erode diversity down to one, which obviously is not a good model of adaptive radiations. In fact, the question of whether diversity impacts adaptive radiations makes no sense in that case, as radiations are just impossible.

However, this is a degenerate property caused by this particular choice of functions (other atypical properties have already been reported for this combination of functions; see e.g. ref 50). As soon as one generalizes the model and relaxes the assumption, as we do in this article, that the two functions have the same Gaussian functional form, this property does not hold anymore. We recover more biologically intuitive predictions, and in particular, two species can evolve to a stable singular coalition and persist on an evolutionary timescale, even if the one-species singular strategy is an ESS. This is also more in line with what is observed in the other two ecological scenarios (body-size and niche) that we studied.

We have added an entire section in the Supplementary Information (Section 4; including six new Figures) in which we discuss this issue and provide detailed evolutionary results for the niche model. As we show, the stable evolutionary coexistence of two species in spite of a one-species ESS is a robust prediction obtained for different choices of functions, not only the Lorentzian $k / \text{Gaussian}$ a case that we retained to illustrate our findings in the main text. The niche model generically has a much richer dynamical repertoire than the Gaussian/Gaussian case (see our bifurcation analyses of the two-species evolutionary dynamics).

Very minor points:

1. 67: why does (1) become a “Gompertz” equation for $\beta=0$?

> We provide below a mathematical proof of this result. The result is reported (though not derived) in the textbook by Goel & Richter-Dyn (1974), cited in the manuscript (line 67).

We start with the proposed growth function $g(z) = (1 - 1/z^\beta)/\beta$, for equation (1).

To reduce clutter z is used to denote the argument to the function (\dot{z} in the article) and b is used in place of parameter β .

At $b=0$ this is an indeterminate form. To compute the limit $\lim_{b \rightarrow 0} g$, we introduce the two functions:

$$u(z) = 1 - 1/z^\beta$$

$$v(z) = \beta$$

$$\text{such that } g(z) = u(z) / v(z)$$

We compute the derivatives of u and v with respect to b as:

$$du/db = d(z^\beta)/db = (z^\beta) \log(z) \text{ which evaluated at } b=0 \text{ simplifies as } \log(z);$$

$$dv/db = 1$$

Finally we use L'Hospital's rule to obtain the limit

$$\lim_{b \rightarrow 0} g(b \rightarrow 0) = du/dv (b=0) / dv/db (b=0) = \log(z)$$

With $b=0$, taking the simple one-species case, we thus get an ecological model of the form $dn/dt = r n \log(k/n)$, which is known as the Gompertz equation for population dynamics.

The resulting form is illustrated below, next to the usual Lotka-Volterra form (dotted curve):

1.128: “A bifurcation from ESS to diversification occurs when, after evolution, one of the species no longer prevents invasion by nearby trait values.” This is a very vague statement. Are you referring to the bifurcation during which an ESS becomes an evolutionary branching point?

> Yes this is exactly what we meant; we have reformulated accordingly (lines 127-128).

Reviewer #3 (Remarks to the Author):

This paper outlines a theoretical analysis suggesting that evolutionary diversification, like that associated with adaptive radiation, is more likely to occur when there is already some diversity present in a region. This hypothesis, which has sometimes been called the ‘diversity begets diversity’ hypothesis, is a controversial one, as it is typically assumed that diversification is most likely to occur when a single species gains access to a wide range of ecological opportunities.

The authors examine a simple, logistic-type model describing the population growth of a single species and how this can be impacted by the presence of other species. They show that, for at least three different ways of describing how competition occurs and impacts population density. The three models of competition are: (1) the ‘niche model’ describes competition for a range of resources; (2) the ‘body size’ model describes competition mediated through body size differences; and, (3) the ‘life history’ model describes a trade-off between competitive ability and colonization rate in a multi-patch environment). Their key result is that, in all three cases, evolutionary diversification in key trait values (resource use, body size, competition-colonization trade-off) is more likely to occur as the number of initial species present increases.

The causes of this relationship I find hard to parse, especially as I am not a theoretician. The authors attribute it to the relative importance of habitat matching, interspecific interactions, and biomass and how these impact so-called ‘evolutionary stability’ as the number of species increases. I’m confused on two issues. The first is ‘evolutionary stability’, which is not formally defined (or if it is, I missed it). I think it means, in the parlance of adaptive dynamics theory, the likelihood that a given population or collection of species can be invaded by a novel type.

> Indeed, “evolutionary stability” refers to the fact of being uninvadable (or unbeatable) by neighbouring variants. It defines whether selection is locally stabilizing (evolutionary stability; genetic variance is not expected to increase) or locally disruptive (evolutionary instability; genetic variance is expected to increase) on the phenotype of the focal species. When there are several species, all species must be evolutionary stable for the community to be evolutionary stable as a

whole. We now provide a definition of evolutionary stability the first time this term is used, and state its implications for diversification (Introduction; lines 45-50) to facilitate comprehension.

The second is the connection between the classic models of adaptive radiation – which sees diversification due to strong resource competition in the presence of abundant ecological opportunity (vacant niche space) is not made explicit. Is the effect of diversity on the propensity to diversify the result of changes to the strength of resource competition or the extent of ecological opportunity, or both? I think it is the latter, given the importance of habitat matching, total biomass, and species interactions, in their selection analysis. But I really am struggling to make sense of this section. To facilitate interpretation it might be useful to cast their models against the standard models of adaptive radiation, to highlight how they differ and the causal factors driving the increased likelihood of diversification.

> The classical adaptive radiation model is not based on an explicit dynamical model of species interactions as ours, but rather on conceptual arguments. It is therefore not obvious to relate the two models at a mathematical level. However, we can make an attempt at this. In our models, ecological opportunity can be considered as constant or, if anything, as decreasing with the number of species, since the distribution of resources (function k) is kept constant, and the distribution of resources probably represents what is intended by ecological opportunity. Competition, on the other hand, varies in different ways, beyond intensity. Schematically, the mathematical components B (total community biomass) and I (frequency-dependent selection on inter-individual interactions) can both be regarded as representing the effect of “competition”. Both B and I vary with diversity, and their variations are involved in the triggering of DDAR (as Figure 5 and S1 show). Our results can thus be seen as a mathematical decomposition of the overall “competition effect”. Simplifying a bit, component B can be said to reflect the total amount of competition (“intensity”), all else equal, whereas component I represents the shape of competition (which also encompasses some aspects of what we'd like to call intensity, but not only). The greatest difficulty in relating to the simple term “effect of competition” is that in multi-species communities, as we consider in this article, “competition” can be either intra-specific or inter-specific, and the two types of competition can have radically different impacts on diversification. Most importantly, in multispecies communities competition can well have an effect that hinders diversification (lines 292-293), whereas it always promotes diversification in the simpler one-species case (which is the case considered in most earlier modeling studies, and probably the one people have in mind when thinking with the classical model of adaptive radiation).

That being said, we have made several efforts in the revised version to relate our findings to the classical adaptive radiation theory:

- First, we now provide the explanations given above on how to relate our results and the notions of ecological opportunity/competition. This is done in the Discussion, where we added nine lines to the paragraph where the causes of DDAR are discussed (lines 300-308). We also added some additional explanations along these lines when discussing potential empirical examples (321-326). We tried to keep it brief to avoid further complicating the presentation of our results, but we think this may help readers familiar with radiation to get a better connection to our findings.
- Second (and this also addresses the reviewer's final concern below), we have added one Figure (Fig. 4) and a corresponding results paragraph (lines 161-170) to study the dynamics of species accumulation through time, together with diversification and extinction rates. These are quantities most commonly discussed with regard to adaptive radiations. We provide a comparison between DDAR and “classical” adaptive radiations, and the comparison is also made in the Discussion (lines 276-283).

My last concern is that I am not entirely convinced that a diversity-begets-diversity scenario is that common in ‘real’ adaptive radiations. While the authors do make an effort to identify a couple of instances where this might occur (in Anoles and in laboratory experiments with *P. fluorescens*) the vast majority of examples in both the real world (see Schluter’s seminal book, *The Ecology of Adaptive Radiation*) and in the lab (consult Kassen’s book, *Experimental Evolution and the Nature of Biodiversity*) suggest that diversity-dependent adaptive radiation is not very common. The authors cite, for example, the paper by Bailey et al (2013) suggesting that competition can sometimes promote adaptive radiation as evidence to support their idea, but I think this is a misreading of their results. Competition did promote adaptive radiation in this system, but only when the competitor could not itself diversify. This effectively means that the strength of resource competition was stronger in the presence of a competitor, only because that competitor was already present (ie – one didn’t have to wait for it to evolve de novo). Perhaps this is the point the authors are trying to make. If so, it would be useful to be clear about this from the start.

> We agree that the evidence available so far suggests that adaptive radiations are predominantly of the classical type. We make this clear in the Discussion and use “atypical” to describe the examples discussed (lines 310-311). Our goal here is just to report the possibility of the DDAR phenomenon, and to suggest it might explain some patterns that may be challenging to explain otherwise.

We improved our discussion of potential empirical illustrations by adding two novel references (Hautmann et al. 2015; ref. 44 and Yoder et al. 2010; ref. 38). Ref. 44, in particular, provides an additional potential example of DDAR from the fossil record. We also significantly rewrote the corresponding paragraph (lines 312-332).

Regarding the results of Bailey et al. (2013), we have expanded our explanations (lines 322-326). The fact that adding a competitor increases the level of competition, and can thus trigger diversification, is indeed consistent with one of our predictions (role of component B). The fact that a competitor might instead exclude the resident strain, or may diversify itself (rather than the resident) is also a possibility in our model. We hope our point is now clearer.

Finally, it might be useful to examine the dynamics of diversification in terms of the number of species through time more formally. Classic models of adaptive radiation predict this dynamic to be an S-shaped curve. How does this model change that prediction? Is the rate of diversification of a focal lineage faster, or is the total number of descendent species higher? Inspection of figure 2 suggests it is both but I would prefer to see a contrast between a classic model of adaptive radiation and these alternative formulations. Such an analysis would facilitate the interpretation of existing macroevolutionary data sets on diversification.

> This is an important point that is quite similar to the first question of Reviewer #2. We addressed it by adding an entirely new Figure (Figure 4) in which we explicitly report the dynamics of lineage diversity over time, and compute the diversification and extinction rates. We study the dynamics of species accumulation, in comparison to classical adaptive radiations, in a novel Results paragraph (lines 161-170) and a rewritten Discussion paragraph (lines 276-287). The main difference between classical adaptive radiations and DDAR is the appearance of an initial lagtime, delaying the burst of diversification. Otherwise, the curves are qualitatively similar. Quantitatively, the equilibrium number of species and the maximum rate of diversification may of course differ, in a specific model, between DDAR and classical radiations, as the two kinds of radiations are predicted to occur for different parameter values (see Figure 4a). Interestingly, when preparing this revision, we found out that this alternative to classical adaptive radiations has been proposed, as a phenomenological model, by Hautmann et al. (2015) based on evidence from the fossil record (ref. 44). Our models can be seen as providing a mechanistic justification to their verbal model.

Despite these extensive comments, I think the model they have studied is intriguing and the results fairly compelling. My comments should be taken in the spirit of trying to improve clarity and to place their work into a context that can be more easily interpreted by those actively studying adaptive radiation in nature or in the lab.

Minor comments:

1. There are a number of odd uses of language throughout that I suspect are translation issues. Some I found: “vanish off” (line 27), “have spent continual efforts” (line 29), “seize this ecological opportunity” (is an anthropomorphism and attributes agency to a diversifying lineage, which is incorrect; line 33). Please revise for clarity.

> We have streamlined the text, correcting the unfortunate formulations.

2. The main point of the paper is only introduced in paragraph 3. Consider reorganizing and condensing paragraphs 1 and 2 to get to the main point faster. The link between diversity and community stability (paragraph 1) really isn't what this paper is about, for example, and is needlessly distracting.

> Following this suggestion we have condensed the first two paragraphs into just one, omitting, in particular, the reference to May's work on the diversity-stability debate (lines 22-26).

3. n_i is introduced on line 101 but not referred to in equation 1, which caused me some confusion. Please clarify.

> We now introduce and define n_i immediately before equation (1) where it first appears (line 62).

REVIEWERS' COMMENTS:

Reviewer #3 (Remarks to the Author):

Calcagno et al have done an admirable job in revising their manuscript and putting their models into context. I cannot see any major flaws in their models, though I reiterate that I am not a theoretician myself and so have not checked their derivations. Instead, my focus is on what their results mean for understanding the process of adaptive radiation and the process of evolutionary diversification.

In this context, I think the leading result of their work – that distinct species can help generate diversifying selection on a focal lineage – does not dramatically change our understanding of how adaptive radiation proceeds. The standard theory is that strong competition (or other forms of ecological interactions generating diversifying selection and promoting diversification) in the presence of ecological opportunity. What Calcagno et al's work illuminates more clearly is that the simplest scenarios involving a single progenitor lineage may not always provide sufficiently strong competition (leading to diversifying selection) to generate diversification. Sometimes other species help this process. This result is certainly consistent with the experimental results by Bailey et al that showed that non-diversifying competitors can promote diversification of a focal lineage by increasing the effective strength of resource competition. It may also explain apparent delays or lags in diversification such as may have happened in the Anole radiation. It remains to be seen how general an effect this is.

I do have concerns with this paper is in some of the choice of language and phrasing. The title, for example, uses the phrase 'diversity-begets-diversity'. I think this is misleading because this phrase is used in the literature to refer more often to the creation of new ecological opportunities by existing species, rather than changes to diversification rates or probabilities as is studied in this paper. When diversity creates ecological opportunities the dynamics of diversification should no longer look saturating but, rather, J-shaped. This is clearly not happening in the models presented here. I suggest using another phrasing that more accurately describes the phenomenon being reported, something like 'diversity spurs diversification' or similar.